# Restoration and Possible Upgrade of a Historical Motorcycle Part Using Powder Bed Fusion

**DOI:** 10.3390/ma15041460

**Published:** 2022-02-16

**Authors:** Lukas Kudrna, Quoc-Phu Ma, Jiri Hajnys, Jakub Mesicek, Radim Halama, Frantisek Fojtik, Lukas Hornacek

**Affiliations:** 1Department of Machine and Industrial Design, Faculty of Mechanical Engineering, VSB-Technical University of Ostrava, 708 00 Ostrava, Czech Republic; lukas.kudrna@vsb.cz; 2Department of Machining, Assembly and Engineering Metrology, Faculty of Mechanical Engineering, VSB-Technical University of Ostrava, 708 00 Ostrava, Czech Republic; jiri.hajnys@vsb.cz (J.H.); jakub.mesicek@vsb.cz (J.M.); 3Department of Applied Mechanics, Faculty of Mechanical Engineering, VSB-Technical University of Ostrava, 708 00 Ostrava, Czech Republic; radim.halama@vsb.cz (R.H.); frantisek.fojtik@vsb.cz (F.F.); 4HORIBA Czech Olomouc Factory, Zeleznicni 512/7, 772 00 Olomouc, Czech Republic; lukas.hornacek@horiba.com

**Keywords:** reverse engineer, powder bed fusion, SS316L, printing simulation, 3D scanning, electronic speckle pattern interferometry, lattice structure

## Abstract

Reverse engineering is the process of creating a digital version of an existing part without any knowledge in advance about the design intent. Due to 3D printing, the reconstructed part can be rapidly fabricated for prototyping or even for practical usage. To showcase this combination, this study presents a workflow on how to restore a motorcycle braking pedal from material SS316L with the Powder Bed Fusion (PBF) technology. Firstly, the CAD model of the original braking pedal was created. Before the actual PBF printing, the braking pedal printing process was simulated to identify the possible imperfections. The printed braking pedal was then subjected to quality control in terms of the shape distortion from its CAD counterpart and strength assessments, conducted both numerically and physically. As a result, the exterior shape of the braking pedal was restored. Additionally, by means of material assessments and physical tests, it was able to prove that the restored pedal was fully functional. Finally, an approach was proposed to optimize the braking pedal with a lattice structure to utilize the advantages the PBF technology offers.

## 1. Introduction

Three-dimensional printing is an additive production technology, in which a physical part is produced layer by layer from a 3D model [1]. It is classified as per the technology used to fabricate the layers, which is governed by the standard ISO/ASTM 52900:2015 [2]. Among various categories, there is the Powder Bed Fusion (PBF) method, which uses a high-energy laser source to bond metal powder particles together layer by layer to form near net shape, fully dense, and functional parts from metals [3].

Three-dimensional printing in general and PBF, in particular, can be deployed to manufacture parts that are not fabricable with the traditional machines, i.e., the lattice structures. By definition, lattices are open-celled structures that are made up of repeated unit cells. A lattice design is specified by its overall dimension, types, and topology (connection) of the unit cells. The most common lattice structure is the strut type, which is composed of strut units connected in a pre-specified manner depending on the loading specifications. By the modification of material usage, topology, and geometry, one can tune the properties of these lattice designs to their needs, some of which cannot be achieved with the bulk materials, e.g., in the fields of mechanics, acoustics, and dielectrics. Indeed, 3D printed structures have been proven to perform significantly better than the cellular structures fabricated with different manufacturing methods given the ability to control the geometry [4]. Thanks to their high stiffness-to-weight ratio, energy-absorbing ability, multi-function, etc., lattices have been adapted in various practical applications including medical implant, transportation, aeronautics, astronautics, etc. [5,6,7,8].

In the field of mechanical engineering, reverse engineering is a process to recreate digital versions of physical mechanical parts [9], as presented in [10,11]. Such a process requires intensive investments in devices and computational facilities, as well as in the designing skills of the designers [12]. Three-dimensional printing has been a promising last piece of the puzzle to fully deliver a reversely engineered product, as it performs excellently in small-scale production of parts with sophisticated geometry. As reviewed in [13], the integration of 3D printing into the reverse engineering process involves first scanning the physical component, then reconstructing the scanned mesh surface. The re-constructed model is then inspected, and its data is analyzed. The completed model is then exported to printable data, which is most often .stl format, and printed with appropriate material and corresponding printing technology. Subsequently, the printed part is subjected to final quality control before its practical deployment.

Nevertheless, as the printing process depends on several parameters of both the printers and the materials, finding an appropriate combination of parameters to deliver a successful print requires a trial-and-failure approach. To minimize such failures or even optimize some printing parameters, some of the software used for printing simulation has been developed and commercialized as listed in [14]. For engineering applications, the attention is mostly paid to either the mechanical, thermal, or both types of simulation in macro-scale. Specifically, given a set of process parameters such as the laser power, scan speed, layer thickness, etc., it is possible to predict the shape distortion of the parts after printing, based on which the designers can distort in advance the original part to compensate for the geometric change. In addition, the potential support failures, the collision between the re-coater and the printed part as the part distorts during printing can be predicted and mitigated.

To nondestructively measure the strength of the parts produced by PBF technology, due to their distinctively rough surface, Electronic Speckle Pattern Interferometry (ESPI) is one of the most suitable methods. In principle, the part to be measured is loaded, and the deformation of the surface regions visible to the measuring camera(s) is measured, which is then calculated to strain and stress [15,16]. The ESPI measurements are non-contact, non-intrusive, applicable for large areas with different materials, insensitive to environmental changes, and with undemanding pre- and post-processing. As reviewed in [17], by varying the optical setups, algorithms, and phase-shift methods, a number of different approaches for 3D ESPI have been realized.

Inspired by the above studies, this paper investigates the restoration of a historic motorcycle braking pedal from the SS316L material with the help of PBF technology. It is anticipated that the original braking pedal was cast. However, due to the small-scale nature of the reverse engineering process, utilization of 3D printing technology is a better option in terms of manufacturing time, not to mention that the technology can offer design freedom, such as the integration of lattice structures into the braking pedal for lighter weight. In particular, the CAD model was first redesigned following the measured dimensions of the original braking pedal. Then, it was subjected to both numerical and physical studies for quality control regarding the shape distortion and strength. Another highlight of this paper is that the braking pedal was optimized with lattice structures without changing its exterior look. During the study, four braking pedal samples were printed, including three solid (one was used for testing the maximum force that a driver can exert on the pedal, one was subjected to 3D scanning and ESPI measurement, one was post-processed), and one optimized version with lattice. The reasons for not combining the 3D scanning, physical tests, and post-processing on one solid pedal would be explained later during the study.

The study was carried out following the workflow illustrated in Figure 1.

Besides, technical notes are listed and discussed within each section and summarized in the conclusions.

## 2. Materials and Methods

### 2.1. Part Design

For the restoration, information related to the braking pedal in the textbook of the motorbike’s manufacturer, Indian Motocycle Co. (now Polaris Industries Inc., Medina, United States), was referred to, as shown below in Figure 2.

The braking pedal numbered S4250 in the company document at its current condition is depicted in the following Figure 3.

The original braking pedal has the outer dimensions of 220 mm × 58 mm × 50 mm. Due to its aging, the braking pedal of the original motorbike is rusty and expected to have its quality downgraded, thus, cannot be subjected to any physical tests for strength. From the manufacturing point of view, it is assumed to be cast and subsequently machined to obtain the hole features.

The material used for the manufacturing of the braking pedal was inspected with an Innov-X Delta mobile XRF spectrometer to have a composition of 98.9–99.3% Fe, 0.7–1.1% Mn. The composition is similar to the S235 JR which has up to 1.5% Mn. However, in this case study, the main goal is to reverse engineer the braking pedal, focusing on modeling the geometry as close as possible to its original counterpart, without any restriction on the material usage, as long as the restored braking pedal is comparatively fully functional.

As aforementioned in the instruction, for the small-scale restoration, the PBF technology was chosen, thanks to the fact that it can fabricate delicate details with ease, within a reasonable amount of time.

### 2.2. Printing Simulation

The printing simulation was carried out before the real printing process to predict and optimize the actual print. Firstly, the CAD model of the braking pedal was imported into Autodesk Netfabb Premium 2020.3 to find the best orientation for printing. The criteria for orientation were: critical angle 46.0°, no support bottom surface, precise volume, smallest rotation between orientations 12.0° (arbitrary), and basic ranking.

For simulation of the printing process, Simufact Additive 2020 FP1 was used. The simulation was performed on the macro scale, meaning that the anisotropic and nonhomogeneous properties of the microstructure were not considered. The main concern at this scale of simulation is the shape distortion, which is taken into account by the inherent strain approach. In particular, similar to a welded structure, a 3D printed part from metal has a total residual strain as follows.
(1)$$\varepsilon_{total}=\varepsilon_{e}+\varepsilon_{p}+\varepsilon_{th}+\varepsilon_{ph\;}$$

The total strain is the sum of elastic strain ($\varepsilon_{e}$), plastic strain ($\varepsilon_{p}$), thermal strain ($\varepsilon_{th}$), and strain induced from the phase transition ($\varepsilon_{ph}$). Accordingly, the inherent strain is defined.
(2)$$\varepsilon^{*}=\varepsilon_{total}-\varepsilon_{e}$$

The inherent strain (ε*) reflects the history of the printing process ($\varepsilon_{p}$, $\varepsilon_{th}$, $\varepsilon_{ph}$) with the shape distortion, after the elastic component ($\varepsilon_{e}$) has been released [19], that is, when the part is cut out from the base plate.

Simufact Additive decomposes the inherent strain (ε*) into three directions for its printing simulation. First of all, the software is calibrated to be as close as possible to reality by means of the cantilevers, as instructed in [20]. From this calibration process, the set of inherent strains is obtained, which is then used to approximate the shrinkage of the printed components in three directions. The obtained inherent strains are listed below in Table 1.

It should be noted that the inherent strains are specific for a particular set of printing parameters. Even if one parameter of the set changes, the calibration must be performed again to obtain the new corresponding set of inherent strains. Thus, the set of printing parameters that are used to print the cantilevers for the calibration purpose and the braking pedal must be the same.

The machine was set to Renishaw—AM400 with a build space of 248 mm × 248 mm × 300 mm. The simulation configuration was Mechanical and the type of simulation was Manufacturing. There are three Manufacturing process stages in total being the Build, Immediate release (of the components and supports from the base plate), and the Support removal. The material of the part was the SS316L_powder from the database. In the Build parameters setting, the layer thickness of 50 µm was used together with the inherent strains in Table 1. The braking pedal was imported to Simufact Additive in the .stl format, with the desired orientation already set in Netfabb in the previous step. The support type was default. The braking pedal and its supports were meshed with voxel (hex elements) sizing of 2 mm, resulting in 19,531 voxels with 43 layers in total.

### 2.3. PBF Printing

As previously mentioned, the Renishaw—AM 400 machine was deployed for manufacturing the braking pedal. The printing parameters that were used are listed below in Table 2.

After printing, the braking pedal was cut out of the base plate and its supports were manually removed, the braking pedal was 3D scanned to inspect the surface distortion in comparison with its CAD counterpart. For the post-process, after support removal, the printed pedal was ground and sandblasted to get rid of the abnormalities on its surface with Cabinet Sandblaster 350 L XH-SBC 350 with S170 steel medium (grain size (355–425) µm). Then, the pedal was tumbled with tumbler OTEC CF1 × 32EL within 120 min in ceramic media DZS 10/10(Otec Company, Pforzheim, Germany).

### 2.4. Three-Dimensional Scanning

The Creaform Handyscan Black 3D scanner (Creaform Inc., Levis, QC, Canada) was used to scan the geometry of the braking pedal then imported to 3Dexperience to compare with the surface deviation result.

### 2.5. Strength Test

Since there is a lack of information on the testing procedures for the integrity of the braking pedal, the author group had to design their own testing procedure. Firstly, a stand was designed from sheet metals with a uniform thickness of 10 mm. The outer dimension of the stand is 275 mm × 150 mm × 90 mm. Its isometric view is shown in Figure 4.

The setup on the left of Figure 4 was used to measure the force that an average rider can exert to the braking pedal. After the force was obtained, the stand, together with the braking pedal, was mounted on a hydraulic machine to test with the same force and then the stresses in critical areas would be measured.

The strain gauge measurement was performed to determine the maximum force with which the braking pedal can be loaded. The test setup and the mounting of strain gauges are shown in Figure 5.

In Figure 5, the braking pedal in the test fixture is loaded with the driver’s foot in the braking position. Two direct strain gauges type LY 3/120 from HBM with temperature compensation for stainless steel were used to measure the deformation in selected places on the braking pedal. One strain gauge was glued on the braking pedal support rod in the longitudinal direction and the second was glued to the rod section, respectively numbered 1 and 2 in Figure 5. The strain gauges were connected to the NI cDAQ 9172 apparatus with a NI 9235 strain gauge card (National Instruments, Austin, United States). The strain was recorded by SignalExpress software from the same company on a personal computer. The sampling frequency was set to 2 kHz.

The braking pedal was loaded three times for the quasi-static and four times for the dynamic test. Subsequently, the stand with the fixed braking pedal was placed in the TESTOMETRIC M500-50CT electro-mechanical machine (The Testometric Company Ltd., Rochdale, United Kingdom) to test for the maximum force that the test driver can exert. Specifically, the braking pedal was loaded as per the setup in Figure 6 to determine the force that corresponds to the maximum measured strain in the previous quasi-static and dynamic tests.

The force would then be used for strength assessment. Before conducting the stress measurement using the Electronic Speckle Pattern Interferometry (ESPI) method, finite element analysis (FEA) must be performed to find the critical areas where ESPI can focus on.

### 2.6. Materials

For the strength simulation, the material properties of the SS316L were studied. In order to get a stress–strain curve and basic material properties the same testing machine was used for tensile testing as for mechanical testing of the pedal).

All specimens were machined from cylinders with a diameter of 10 mm printed in a horizontal direction according to the scheme of the geometry shown in Figure 7. A semi-automatic extensometer with a strain gauge length of 25 mm was used to measure the longitudinal strain.

The strain rate sensitivity of additively manufactured SS316L under room temperature is significantly lower than that of the conventional one as reported elsewhere [21]. However, it was reported just on results from the low-cycle tests in the study mentioned above. Thus, new tensile tests were realized on specimens made of a virgin powder here under three different strain rates 0.1; 0.5; 1%/s (corresponding to the position rate of 3.5; 17.5; 35 mm/min). The yield and ultimate strength dependency on strain rate obtained on specimens are shown in Figure 8.

Microstructural investigations and a comparison of mechanical properties of vertically/horizontally printed specimens with the conventional ones can be found in [22,23], respectively.

A comparison of the monotonic tensile curve of both additively manufactured variants is presented in Figure 9. The stress–strain curves were obtained under the strain rate of 0.29%/s (position rate 10 mm/min).

In Table 3 below, there are listed two variants of the SS316L materials, recycled and virgin powder, which were evaluated from the tensile tests corresponding to the strain rate of 0.29%/s.

A prototype of the braking pedal for subsequent measurements was 3D printed with the recycled powder of SS316L in three pieces.

The same elastic properties, Young’s modulus 204 GPa and Poisson’s ratio 0.29, were used for all the simulations and measurements in this work. Elastostatic analysis neglecting inertia effects was performed. As for the boundary conditions, the holes to be mounted to the stand were fixed at six degrees of freedom (DOFs). The maximum force measured in the previous step was calculated to pressure and applied on the surface where the foot is placed. These boundary conditions are shown in Figure 10.

After the FEA, the critical areas on the braking pedal with high stress levels are found. Notably, identifying the critical areas is very important for ESPI stress measurement since high accuracy of strain measurement can only be obtained if the zone to be measured is isolated correctly.

The stress level at the hot spot of the pedal was measured with the help of ESPI. For ESPI, the full-field measurement was performed using Dantec Dynamics Q100 equipment, which is essentially a unique device for fast, non-destructive measurement of strain and stress fields on a component without difficult component preparation or marking. The measured area is illuminated by a laser beam from four different directions and the scattered light is recorded using a central CCD chip. In addition, if a 3D geometry is recorded, it enables to automatically quantify deformations and strains on the 3D surface. With known material properties, such as Young’s modulus and Poisson’s ratio, the stress distribution can also be evaluated in contours.

### 2.7. Optimization with Lattices

Due to the advantages the PBF technology offers in fabricating parts with complex geometry, the braking pedal can be forwarded for optimization study to redesign with lattices. The aim of the optimization is to save weight by carving out the material from the original part, or, in another word, to distribute the material by means of column lattices where it is needed the most, with regard to a set of predefined criteria.

For optimization, it is necessary to anticipate some scenarios where the part to be optimized is loaded the most, to prepare the corresponding loading cases, that is, under the maximum force obtained from Section 2.5 The Altair Inspire software, version 2020.1.1, was used for the lattice structure redesign. First, the strength of the braking pedal under the 815 N force was examined. The boundary conditions were as previously shown and the material properties were taken in Table 3. Then, given the FEA results, it was possible to establish the design space, which was to be optimized with lattices. Subsequently, some information related to the lattice topology and constraints were described for the redesigning with lattice. It is necessary to edit the run with a number of different combinations of the optimizing parameters to obtain the best design, which is later reported.

## 3. Results and Discussion

### 3.1. Part Design

Based on the geometric and material inspection as well as the plan for manufacturing, the CAD model of the braking pedal was redesigned as follows in Figure 11.

The model at this stage was ready for the subsequent PBF printing simulation.

### 3.2. Printing Simulation

Given the aforementioned criteria for creating the supports, Netfabb randomized a number of possible orientations and corresponding supports as depicted in the below Figure 12.

The numbering from (1) to (15) in Figure 12 indicates the ranks of different orientations for printing based on a set of criteria, which are shown in Figure 13.

The color scale in Figure 13 goes from green (the best) to yellow (acceptable) then to red (the worst) rank. As can be observed in Figure 12, all the vertically or diagonally oriented options increase the change of insufficient cooling, which may lead to severe part distortion. For the rest, the deciding criteria were to reduce the post-processing efforts spent on the support removal by minimizing the support area, as well as to save the material usage by fabricating as little support volume as possible. Additionally, the orientation of the part in PBF affects the surface topography formation and post processes [24], and if the print is oriented parallel to the base plate, it is possible to obtain lower roughness in comparison with other orientations [25]. As for the strength of the printed pedal, if the layer effects and the way the braking pedal is loaded during operation are considered, the vertically oriented one such as option (4) is the most promising candidate. However, since option (4) is high, it will take more time for printing, the slender and high supports will not ensure sufficient cooling leading to the shape distortion, and yet there is the surface roughness factor to be considered.

In view of these constraints, the two ideal options are (2) and (3), which have the best printability without the need to considerably compensate the strength. Subsequently, (2) was chosen and proceeded for the next steps, anticipated to have better surface roughness, despite having a higher amount of support volume compared to (3).

After determining the optimal printing direction, the pre-oriented model is exported to Simufact Additive for the simulation of the printing process. It is worth recalling that the inherent strains in Table 1 are among the most important factors for the calculation of the shape distortion. The potential surface deviation from the CAD model of the braking pedal was predicted to be as follows in Figure 14.

It can be observed that in comparison with the original shape, the cylinder was bent upward in two ends. This was also indicated by the slight distortion of the bottom surface of the rod section. The overall deformation of the braking pedal was slightly tilted because, in option (2), it was rotated at a small angle. The maximum distortion was 0.76 mm at the pedal end and the minimum distortion was −0.81 mm at the cylinder end.

### 3.3. PBF Printing

Since it was possible to print only one pedal on a base plate at a time, there were three prints of the solid pedals in total. After printing, the three pedals were removed from the base plates together with their supports, as in Figure 15.

The post-processing of the printed pedals in this study included cutting of the pedal from the base plate by saw, support removal, surface treatment, and additional machining to obtain the hole features. In order to compare the printed pedal with its CAD counterpart, surface treatment and machining were not realized at this stage.

The first print was to verify the printing setup. Since the pedal resulting from this first print did not meet the quality control requirements, it was subjected to surface finishing for demonstration purposes. The second and the third pedals were two successful prints, one of which was used for testing the maximum force from the driver. The other one was subjected to geometrical inspection with a 3D scanner then ESPI strength assessment. It is noteworthy that the surface finishing was not conducted on the second and third solid pedal. This was because the second pedal was tested to fail and the rough surfaces of the third pedal must be maintained as in Figure 15 so that the ESPI system can operate accurately.

### 3.4. Three-Dimensional Scanning

The surface deviation results of the second solid pedal in comparison with its CAD counterpart can be seen in Figure 16.

With regard to Figure 14, it can be concluded that Simufact Additive can anticipate the hot spots to some degree of accuracy. However, the maximum and minimum distortion were almost double the predicted ones. This could be because of inaccurate inherent strains, or the many factors in practice ranging from machine parameters, powder, order of support removal, etc. Remarkably, one can see that the bottom side of the braking pedal was with rough and pointy finishing, remaining after the support removal. To obtain a better bottom surface finishing, it should be additionally ground with sandpaper, grinder, or more thoroughly by combining the two methods with tumbling [26]. Figure 17 depicts the first solid pedal after surface finishing to demonstrate the possible surface roughness improvement.

In Figure 17, a crack-like region can be noticed in the bottom surface of the rod section due to the lack of powder. As for surface finishing, the pedal was ground, sandblasted, and tumbled for two hours using the equipment as aforementioned. As a result, shiny and better surface finishing was obtained, while the abnormalities on the bottom surface of the rod section were effectively removed (see Figure 16).

### 3.5. Strength Test

Figure 18 plots the deformation curves from the strain gauges 1 and 2 in Figure 5, being pressed under the maximum quasi-static and dynamic force that an average driver can exert on the pedal when sitting on the motorcycle.

The maximum strain for these two tests amounted up to approximately 1345 µS. After the test in Figure 6, the corresponding force for the maximum measured strain was determined to be approximately 815 N.

For strength simulation, the force of 815 N was applied as pressure on the surface of the pedal end where the foot is put on. The maximum Von Mises stress acting on the pedal under the force of 815 N is approximately 481 MPa. As it is above the yield strength of the material considering recycled powder, it may cause some difficulties for the ESPI measurement. Thus, the force for FEA and ESPI test was reduced to 200 N. Within the elastic region, it is reasonable to assume that after obtaining the ESPI results for 200 N, the results for 815 N can be roughly calculated by multiplying the 200 N by a factor of 4, according to Hooke’s law.

In this subsection, only the Von Mises stress distribution of 200 N was depicted to compare with the ESPI results, as can be seen in the below Figure 19.

The most critical area was the radius with the Von Mises stress level amounted up to 118 MPa. After simulation for the weak spots, the printed pedal was subjected to the strength test with ESPI system. It was mounted on the stand and tested with 200 N force following the setup in Figure 6. The stress contours were calculated from the material properties given in Table 3. Accordingly, the Von Mises stress contours are shown in Figure 20.

The most critical area isolated by the blue rectangle had the maximum equivalent Von Mises stress of approximately 132 MPa. Compared to the FEA results, the difference was approximately 11% (118 MPa in simulation and 132 MPa in ESPI). The stress distribution was predicted well with the FEA.

### 3.6. Optimization with Lattices

Since the pedal under operation was analogous to a cantilever subjected dominantly to bending moment, the bending stress should be 0 on the neutral axis and linearly grew to its extreme in the outermost fibers on the top and bottom of the pedal’s rod section. This was confirmed by the Von Mises stress result in Figure 18. Therefore, there was a space for optimization by removing the inner material of the rod section and replacing it with a collection of lattice structures, which is discussed in detail below.

It should be noted that the Von Mises stress and deformation values presented henceforward were solely from the simulations under the maximum force of 815 N. The aim of these simulations was to investigate theoretically the maximum Von Mises stress level the original and optimized pedals would experience in practice. Furthermore, as previously discussed, under the maximum load of 815 N in the physical test, the resulting Von Mises stress value was 481 MPa. This was approximately over the yield limit of 467 MPa of the recycled powder given in Table 3 and would result in problems with the ESPI measurement.

As for the simulations of both the solid and optimized pedal with lattice structures, the material properties in Table 3 were used. The Von Mises stress distribution is as follows in Figure 21.

The most critical area was the radius, which was the same as in Figure 19 with a higher level of stress, about 428 MPa. This was approximately 12% different from the FEA result in Inventor.

Following the strength assessment, the area to be optimized was wrapped with a design space as shown in Figure 22.

After certain iterations, the best set of criteria was found, that was, lattices’ target length 5.5 mm, maximum/minimum diameter parameter 0.3 mm, fill the design space with 100% lattice, and mass target of 15% remaining from the total volume of the design space. The best optimized result that was obtained from this setup is depicted in Figure 23.

It can be observed that the distribution of lattices was denser at the critical radius area. On the other hand, the area where the foot would rest was the least loaded, thus, was not so densely filled with lattice structures. It should be noted that the design space was linked to the big hole so that the excess powder inside the lattice space can be removed after printing. Powder removal was the reason why the full geometrical features for the optimized version remained.

The Von Mises stress distribution of the optimized pedal is in the following Figure 24.

The maximum stress was reduced to 421 MPa and the rest was redistributed to the lattices near the bottom fiber of the pedal. The Von Mises stress level in the lattices can be up to 106 MPa.

Subsequently, Table 4 lists the comparison between the original and the optimized pedal.

As aforementioned, the comparison in Table 4 is only for numerical reference. Thanks to optimization, it was possible to save 16% of the mass. The maximum Von Mises stress was reduced by 8% and redistributed to the lattices. Since the material was removed, the stiffness of the optimized pedal was not the same as the original, resulting in the maximum tip deformation being 12% higher.

In comparison with the original pedal, the redesigned lattice structure has a lighter weight and reasonably higher max deformation. Since the force applied throughout the entire study was the maximum force that the driver can exert, it can be assumed that there would be lower force during the normal operation, thus, 1.3 to 1.5 safety factor calculated from the yield limit could be expected. However, it must be subject to additional tests for strength and fatigue tests keeping in mind the knowledge about the material properties of the 3D printed SS316L [23,27,28], and the lattice structures [29,30,31]. The non-symmetric cyclic loading naturally applied to the braking pedal could cause the accumulation of plastic deformation in the hot spot area for high levels of load. However, as shown in the previous study [28], uniaxial ratcheting under maximal stress of 500 MPa and minimal stress of −50 MPa vanishes and leads to a plastic shake-down. That was why it can be concluded that the braking pedal fabricated with the virgin powder was with better reliability.

As a step further, the optimized pedal was printed and shown in Figure 25.

There was an attempt to print the optimized pedal with full geometrical features for testing. However, the print did not pass the quality control process, since there was a lack of material at the unfinished circular end of the pedal. This problem was caused by the insufficient number of supports at the zone. Thus, it was decided that the strength test would not be conducted on the optimized pedal. Instead, it would be cut with the electrical discharge machining (EDM) method to investigate the internal lattice structures. The cut surfaces were of a black color because of the material burn under EDM cutting. In this print, the hole features were printed, however, with polygon shapes instead of circular. This is due to the coarse mesh that was used in the .stl file. There was no problem with the shoulder feature near the small hole. Besides, there were droplets of powder that were not melted thoroughly in the holes and on the bottom surface of the rod section, where there were supports. These problems can be avoided by suppressing the features before printing and applying machining in the post-process. The droplet in the bottom surface of the rod section can be eliminated by grinding. In addition, it is possible to obtain a smoother and shiny overall surface by the tumbling process described in [26].

The printing of this optimized pedal case demonstrates the fact that in order to deliver a successful print, there are a number of factors to be considered, in terms of the machine operation, powder usage, supports planning, etc. Besides, printing with full geometrical features was not an ideal option for parts that require precision. In practice, with the help of computers, it is possible to minimize but not eliminate the causes leading to the unsuccessful print, thus, the hands-on experience of the machine operators and the trial-and-failure approach is needed.

## 4. Conclusions

In summary, the process of how to reversely engineer a historical braking pedal of a motorcycle was reported. The contributions of this paper can be summed up as follows:A framework proposal for reverse engineering a historical part, that is, the motorcycle braking pedal using the PBF technology.The braking pedals printed by the SS316L were studied from both the material and the geometry perspectives.Additively printed SS316L was newly investigated from the strain rate sensitivity point of view by tensile tests (a virgin powder).The printing process simulation of the pedal was conducted to determine possible failures that would occur in terms of supports and geometry distortion.One of the solid pedals was subjected to geometry inspection using 3D scanning, and strength inspection using both the FE and physical tests.As a highlight of this paper, the design optimization of the braking pedal with lattice structure was carried out keeping in mind the constraints of the restoration work.The optimized pedal was printed and cut to showcase the internal lattice structures and the problems that could happen with 3D printing using PBF technology.

The aim of reverse engineering the historical braking pedal using the PBF technology was achieved. Specifically, the shape of the original pedal was successfully restored and proven to be fully functional, as a result of material studies and physical tests. Furthermore, a potential approach was proposed to upgrade the braking pedal in terms of weight saving with lattice structures. Notably, the material selection, including powder usage, and its properties, associated with specific printing parameters, must be considered carefully. For future studies, the application of 3D printing in the field of reverse engineering can be further investigated in terms of material properties, the strength of printed parts, designs with topology optimization, etc., on more sophisticated designs operating in more extreme environments.

## Figures and Tables

**Figure 1 materials-15-01460-f001:**
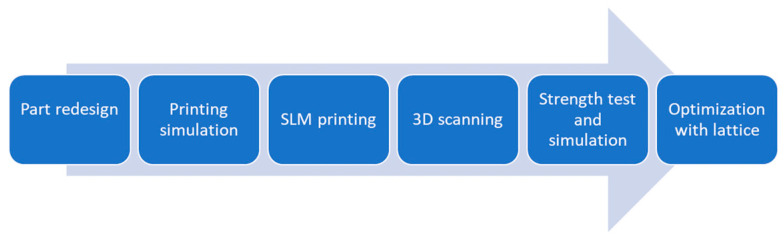
Workflow of the restoration case study.

**Figure 2 materials-15-01460-f002:**
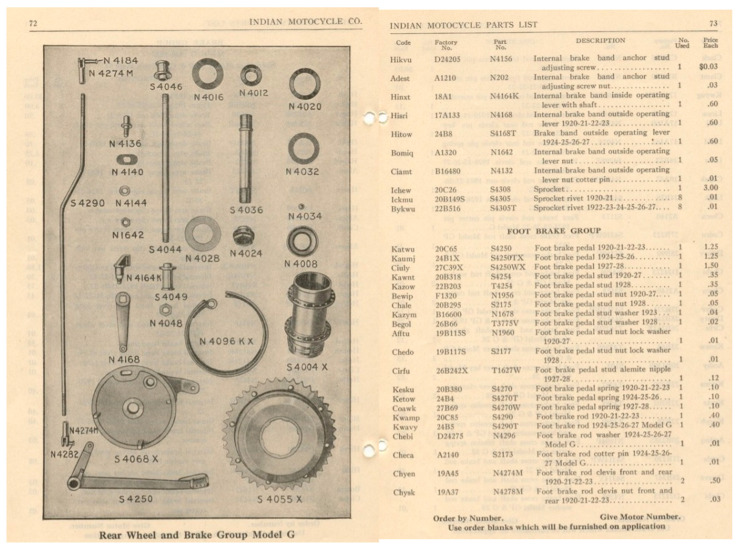
Information on the braking pedal in the document [18].

**Figure 3 materials-15-01460-f003:**
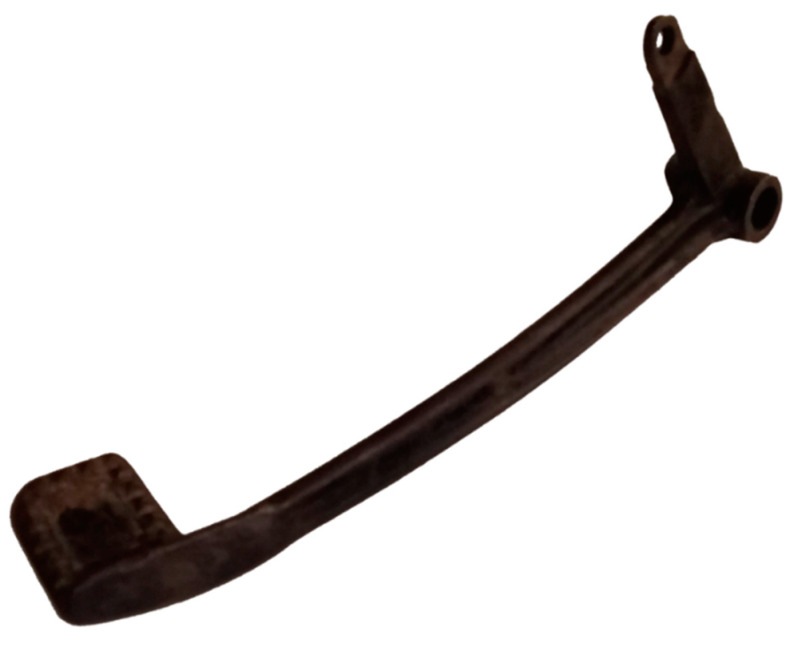
The original braking pedal.

**Figure 4 materials-15-01460-f004:**
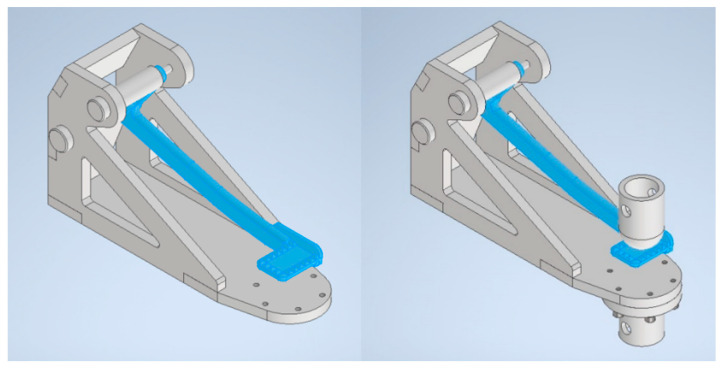
Setup of the braking pedal and the stand and on the hydraulic machine.

**Figure 5 materials-15-01460-f005:**
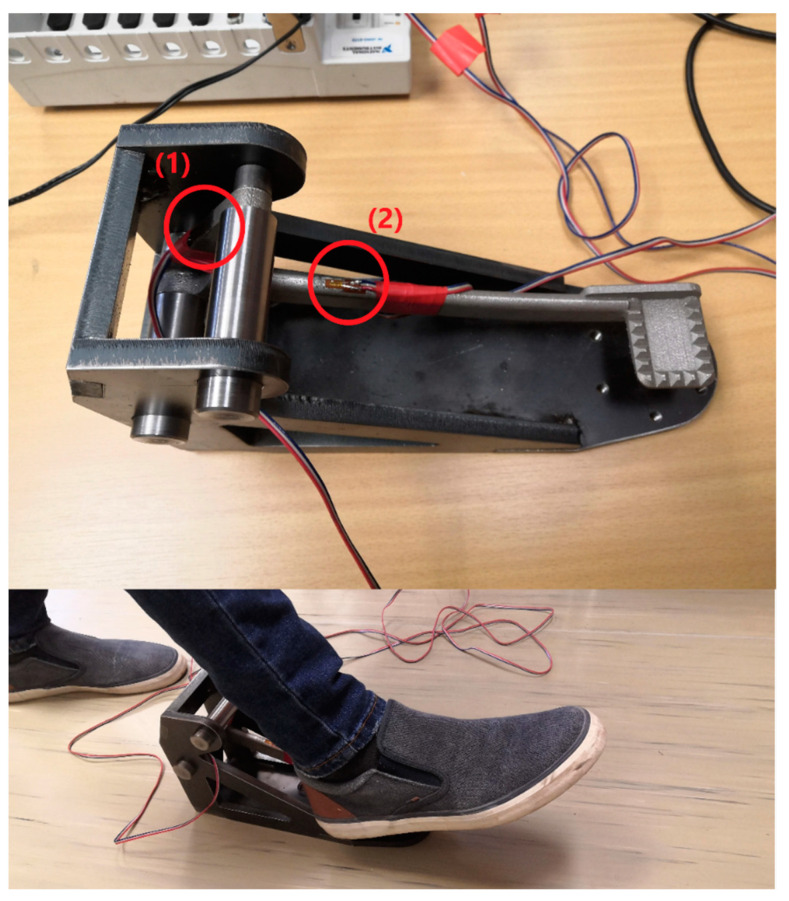
Setup for strain gauge measurement.

**Figure 6 materials-15-01460-f006:**
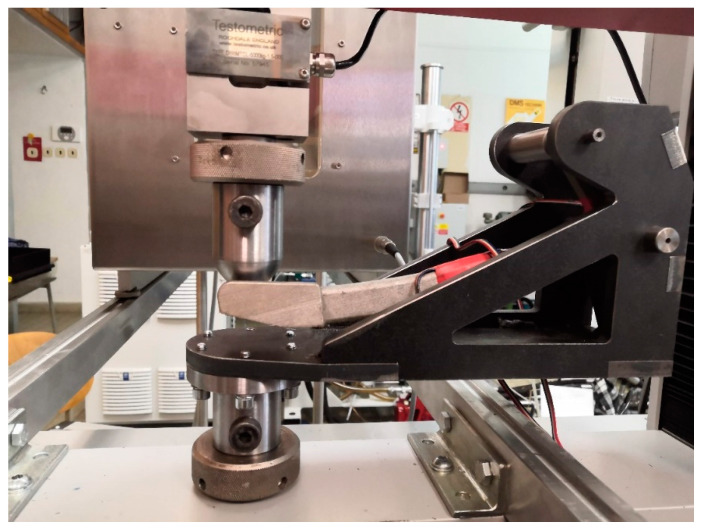
Setup for force measurement.

**Figure 7 materials-15-01460-f007:**
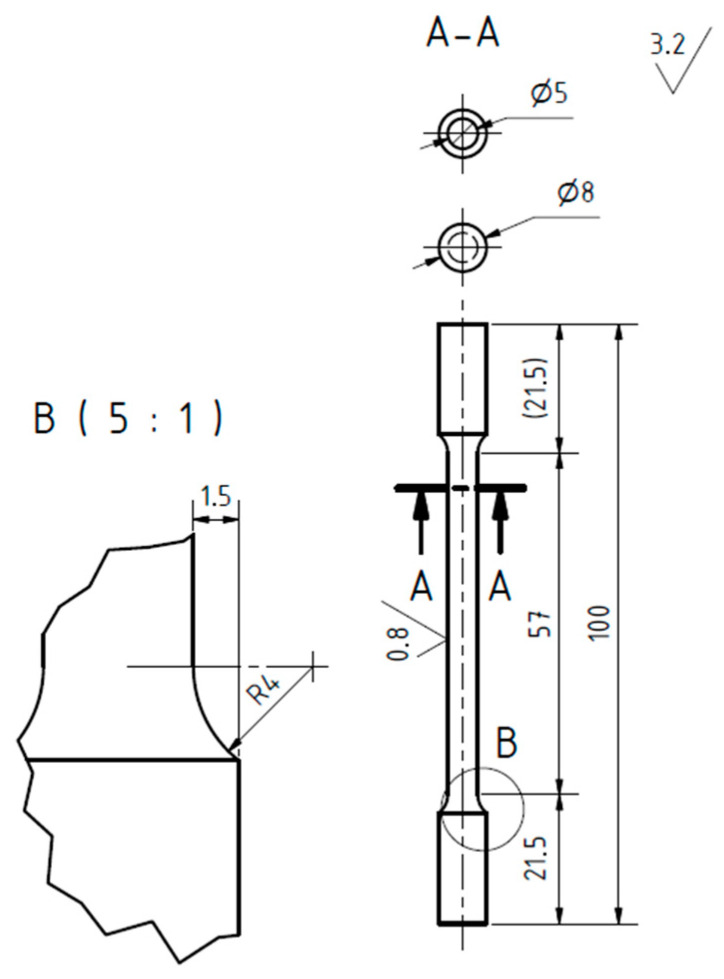
A scheme of specimen geometry for tensile testing.

**Figure 8 materials-15-01460-f008:**
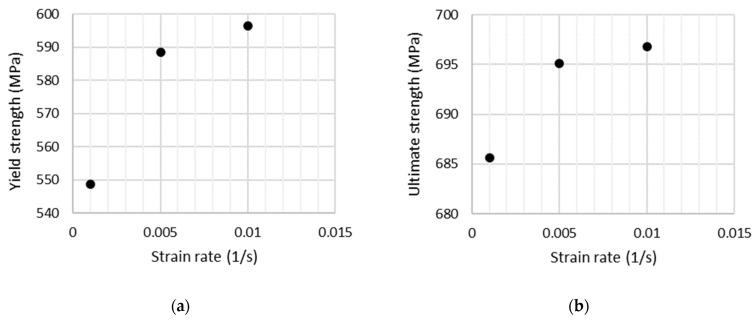
Strain rate sensitivity of SS316L printed in the horizontal direction with a virgin powder: (**a**) yield strength, (**b**) ultimate strength.

**Figure 9 materials-15-01460-f009:**
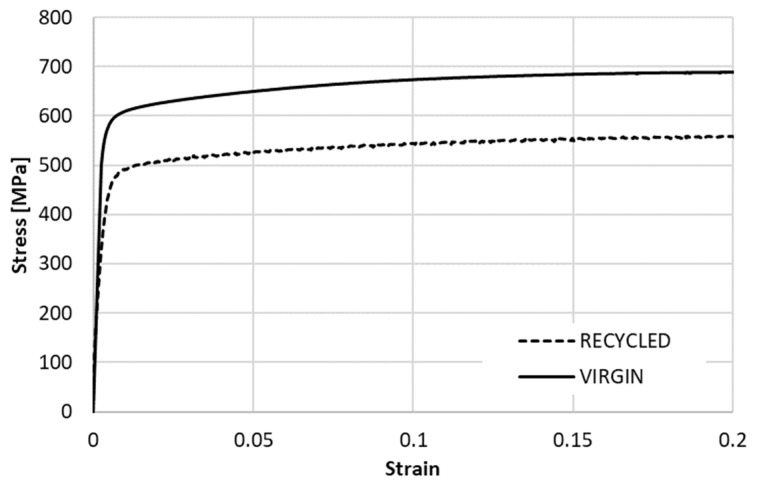
Monotonic stress–strain curves obtained for SS316L.

**Figure 10 materials-15-01460-f010:**
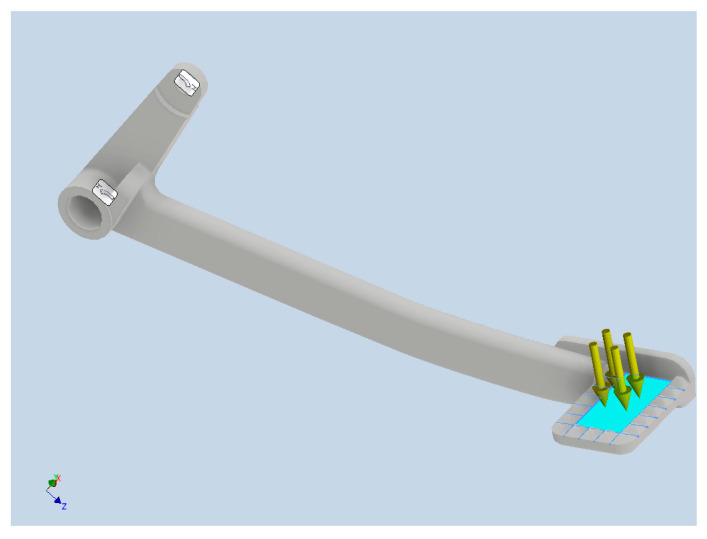
Boundary conditions for the strength simulation.

**Figure 11 materials-15-01460-f011:**
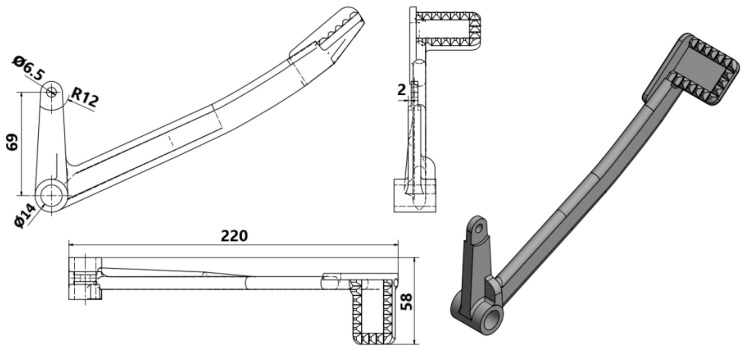
Technical drawing of the reversely engineered braking pedal.

**Figure 12 materials-15-01460-f012:**
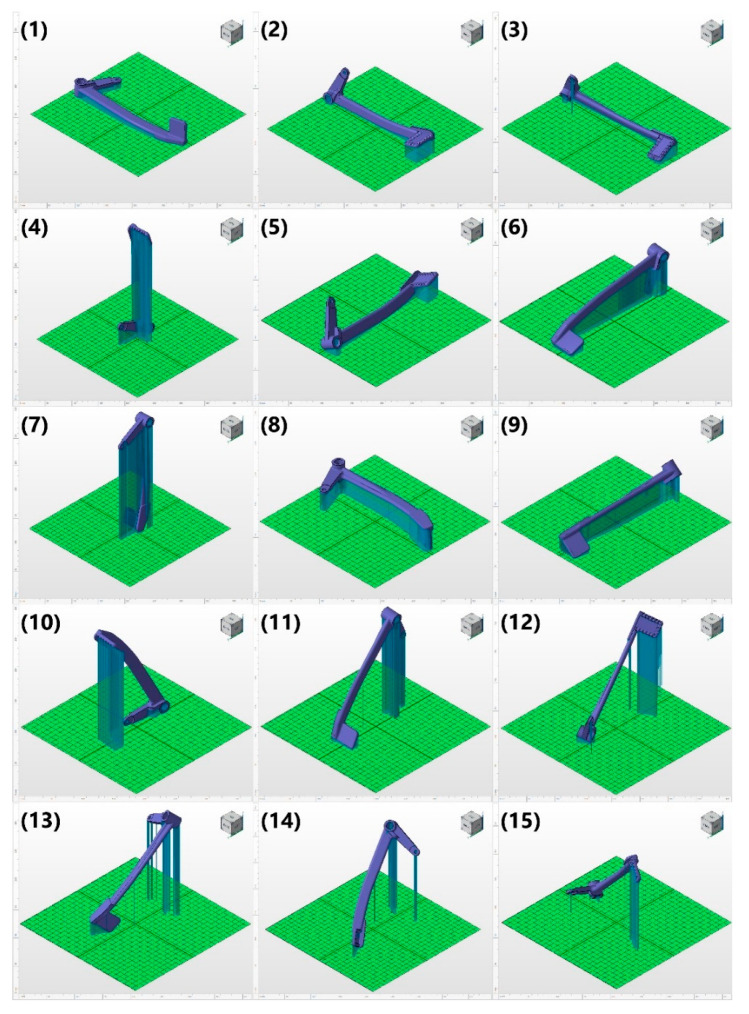
Possible orientation of the braking pedal on the base plate.

**Figure 13 materials-15-01460-f013:**
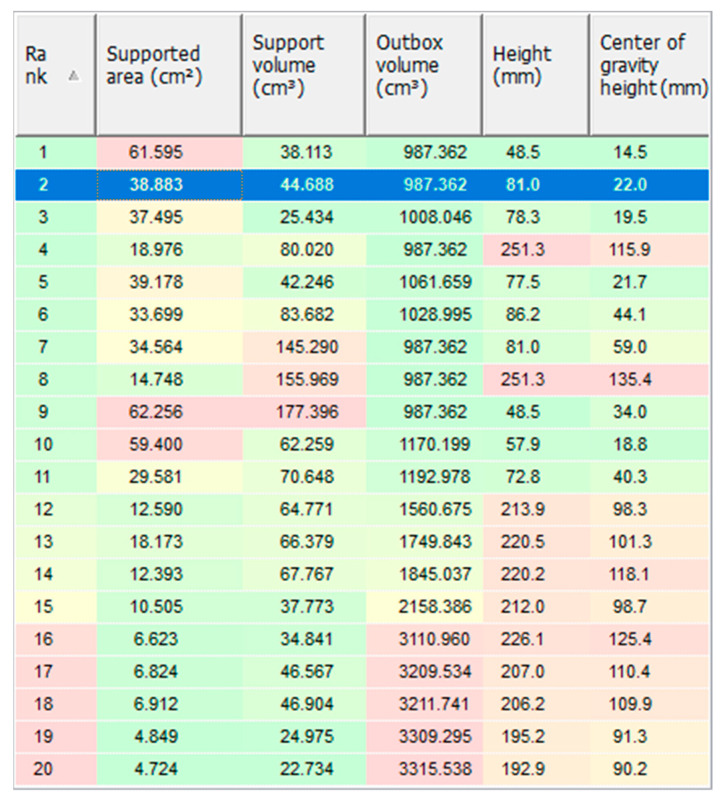
Possible orientation of the braking pedal on the base plate with the selected option highlighted.

**Figure 14 materials-15-01460-f014:**
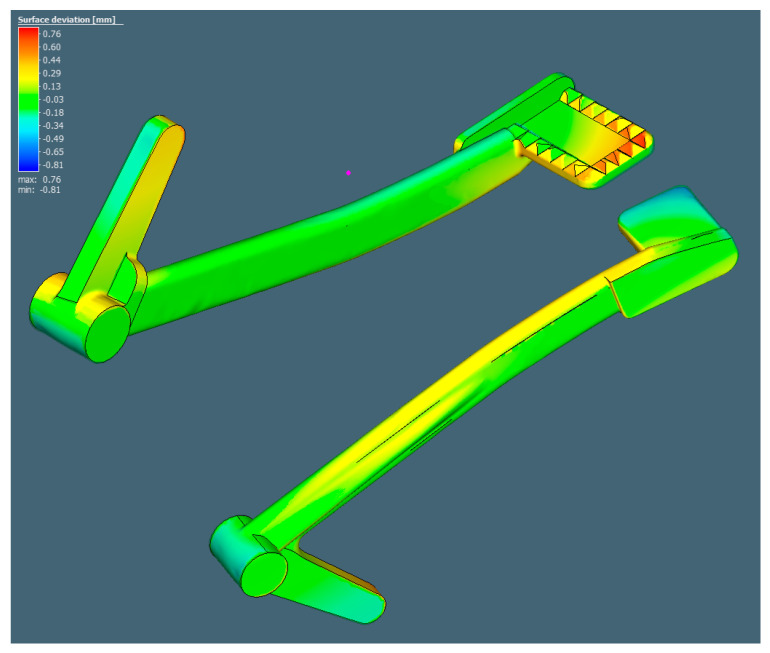
Surface deviation prediction in Simufact Additive.

**Figure 15 materials-15-01460-f015:**
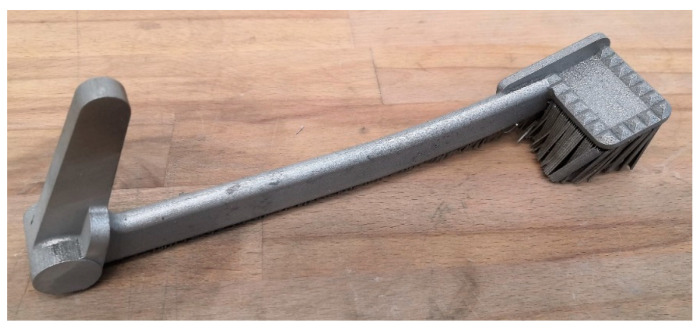
The printed braking pedal.

**Figure 16 materials-15-01460-f016:**
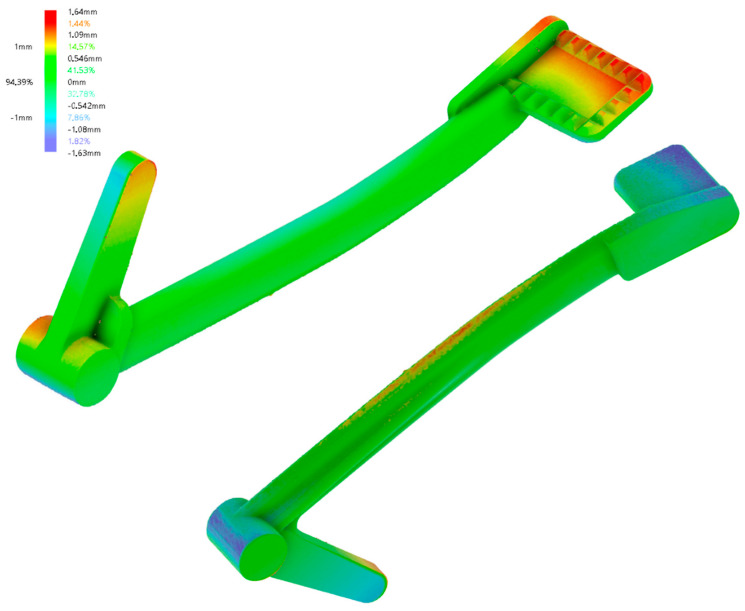
Surface deviation measured with a 3D scanner.

**Figure 17 materials-15-01460-f017:**
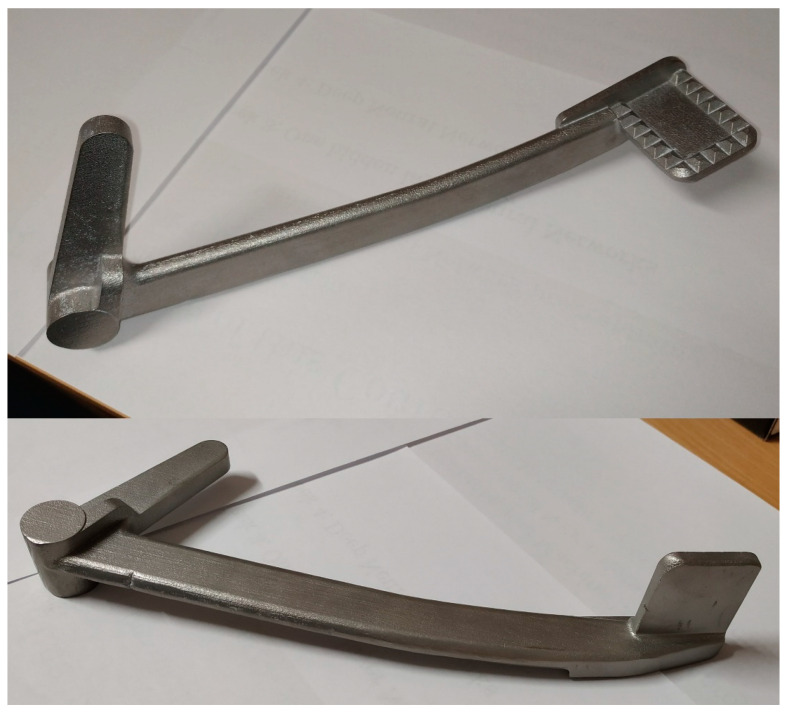
Surface finishing with grinding, sandblasting, and tumbling.

**Figure 18 materials-15-01460-f018:**
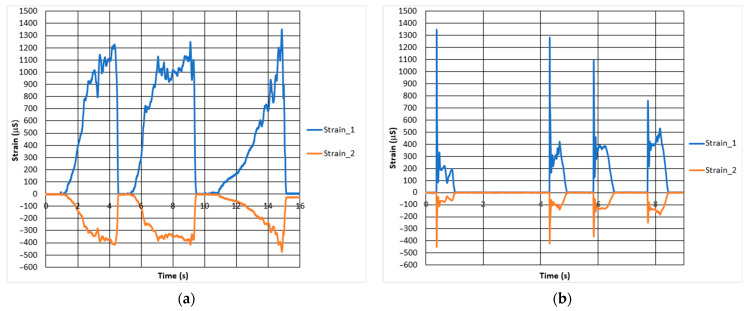
Results for strain measurement (**a**) quasi-static case; (**b**) dynamic case.

**Figure 19 materials-15-01460-f019:**
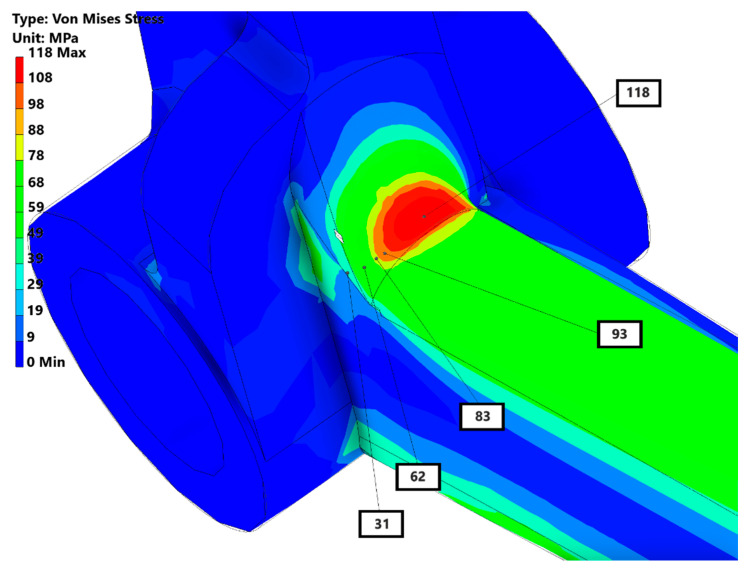
Von Mises stress results in Inventor.

**Figure 20 materials-15-01460-f020:**
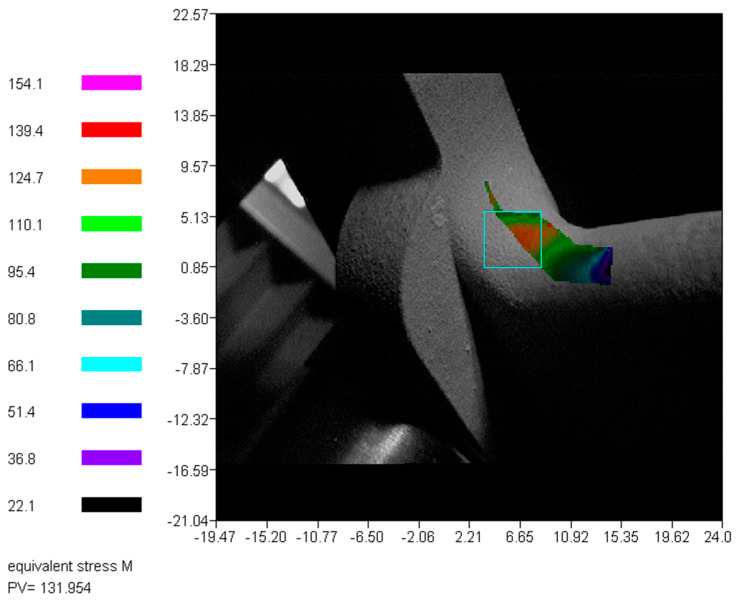
Von Mises stress result obtained with ESPI method.

**Figure 21 materials-15-01460-f021:**
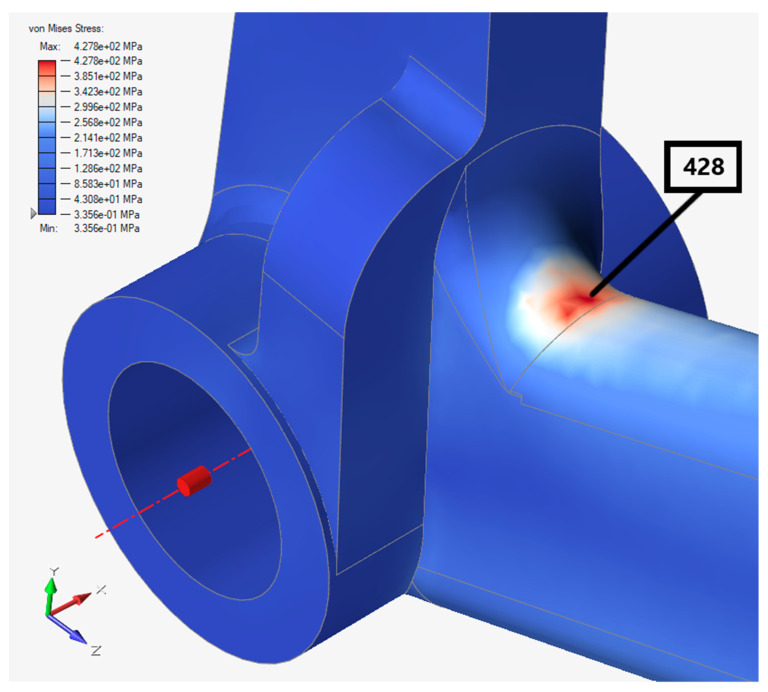
Von Mises stress result in Altair Inspire.

**Figure 22 materials-15-01460-f022:**
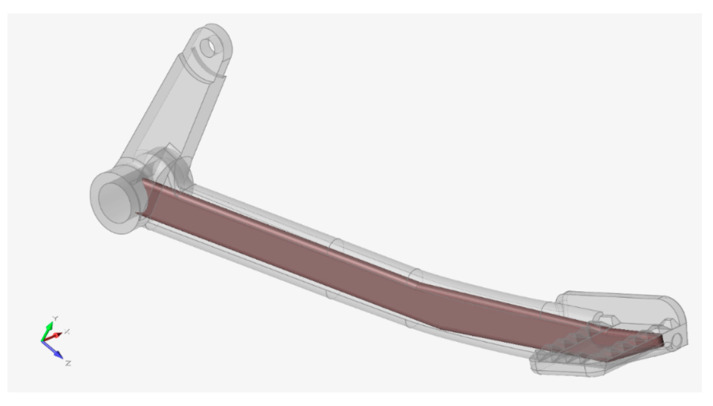
Design space.

**Figure 23 materials-15-01460-f023:**
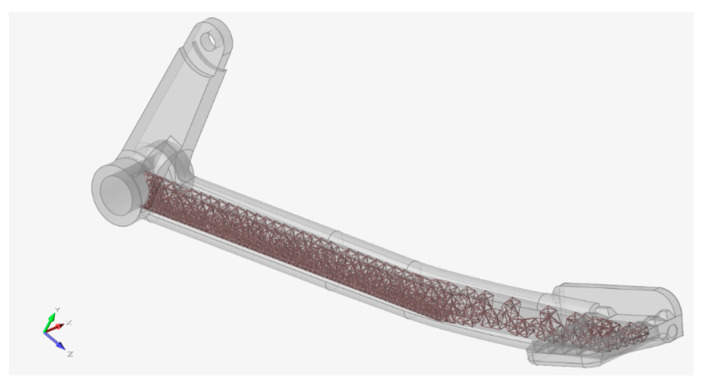
Design space optimized with lattices.

**Figure 24 materials-15-01460-f024:**
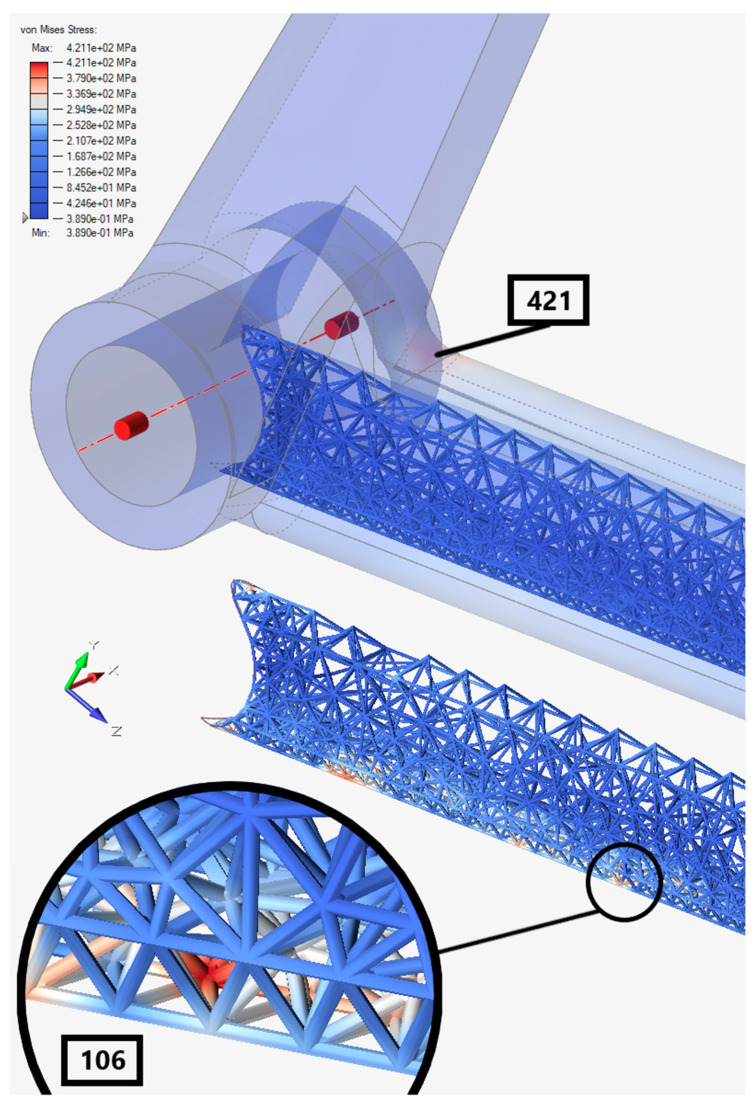
Von Mises stress result on the optimized braking pedal.

**Figure 25 materials-15-01460-f025:**
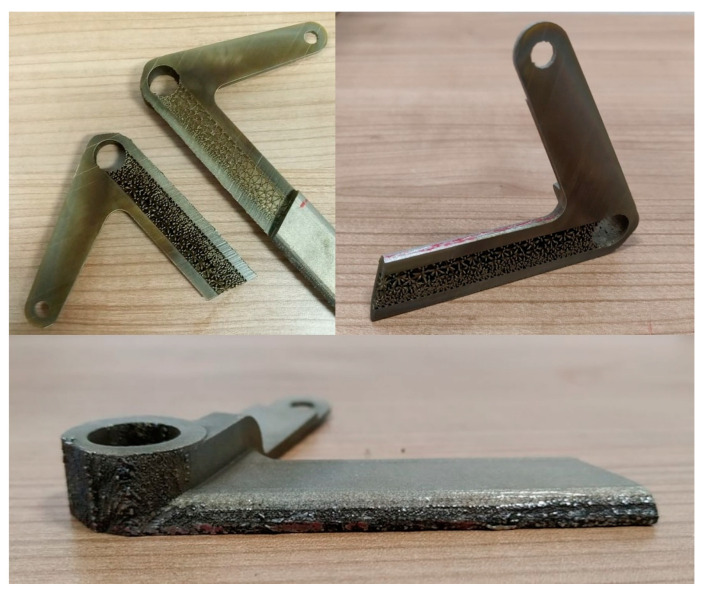
The cut optimized pedal for view into the internal lattice structures.

**Table 1 materials-15-01460-t001:** Inherent strains calibrated in Simufact Additive.

Inherent Strain	Value
Ɛ_xx_	−0.00286296
Ɛ_yy_	−0.00277407
Ɛ_zz_	−0.03

**Table 2 materials-15-01460-t002:** Printing parameters.

Parameter	Value
Laser power	200 W
Scan speed	650 mm/s
Layer thickness	50 μm
Hatch spacing	0.11 mm
Increment rotating angle	67°
Preheat temperature	Ambient

**Table 3 materials-15-01460-t003:** Material properties of the 3D printed SS316L.

Property	Recycled Powder	Virgin Powder
Young’s modulus	204 GPa	200 GPa
Poisson’s ratio	0.29	0.29
Yield strength	467 MPa	572 MPa
Ultimate strength	614 MPa	691 MPa

**Table 4 materials-15-01460-t004:** Comparison between the original and the optimized braking pedal in simulation.

	Weight (kg)	Volume (mm^3^)	Von Mises Stress (MPa)	Max Deformation (mm)
Original	0.52344	65,496	459	2.67
Optimized	0.44033	55,044	421	2.99
%	−16	−16	−8	+12

## Data Availability

All data contained within the article.

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
