# Peer review of "Restoration and Possible Upgrade of a Historical Motorcycle Part Using Powder Bed Fusion"

_materials, 2022, doi:10.3390/ma15041460_

Round 1
Reviewer 1 Report
This research describes a good engineering case about the application of selective laser melting (SLM) on braking pedal. Its main content presents a workflow how to restore a motorcycle braking pedal from material SS316L with the SLM technology, which has a good guiding effect. However, this paper lacks in-depth research value. There are no any contrast experiments on braking pedal between traditional processing technology and SLM technology. It is a good idea to treat some parts of the pedal with lattice structure, but there is no experiment and in-depth study. There are also errors in the typesetting of the paper. In addition, the research should contain the section of discussion.
Author Response
Thank you for your comments, which have helped us to improve the paper considerably. We noted our answers to your comments as follows:
This research describes a good engineering case about the application of selective laser melting (SLM) on braking pedal. Its main content presents a workflow how to restore a motorcycle braking pedal from material SS316L with the SLM technology, which has a good guiding effect.
However, this paper lacks in-depth research value. There are no any contrast experiments on braking pedal between traditional processing technology and SLM technology.
- We chose the experiments which could be applied for the traditionally manufactured pedal to apply on the SLM printed pedal so that we can have a reference on how different they are. We also considered the typical characteristics of the 3D printing technologies considering the anisotropic properties of the material, the differences between the virgin and the recycled powder, the layer effects when orienting the build in the printing chamber, the surface finishing, etc., which are not applicable for the traditionally manufactured pedal.
It is a good idea to treat some parts of the pedal with lattice structure, but there is no experiment and in-depth study.
- We printed the optimized pedal. The print was unsuccessful. However, we decided to keep the results and discussed about the possible failures which could happen when printing with SLM as another practical case study. Due to the time and resource constraints for the project, we cannot continue with other printing and testing.
There are also errors in the typesetting of the paper.
- Reviewed.
In addition, the research should contain the section of discussion.
- Discussion is combined with Results and further elaborated.
Besides, we changed the term Selective Laser Melting (SLM) to Powder Bed Fusion (PBF) to be more accurate following the standard ISO/ASTM 52900:2015.
Reviewer 2 Report
The use of SLM in AM is becoming more ubiquitous. This work provides another fine example (using reverse engineering) of its applicability. While the results/outcomes appear reasonable, there could be various improvements.
Concerning the presentation, much of the work describing the process parameters, materials, etc. could be condensed into one or two tables. There appears to be an exorbitant number of "minor" sections that could be reasonably combined to form a few larger sections.
In terms of technical content, while the experimental and computational tools are described, the level of detail is sparse in many instances. Reproducibility by an outside practitioner would be very difficult, particularly in terms of the computational work. What governing equations were used, what were the initial/boundary conditions, what simplifying assumptions were used, etc.
Author Response
Thank you for your encouraging comments. We have improved our paper accordingly and noted below our answers:
The use of SLM in AM is becoming more ubiquitous. This work provides another fine example (using reverse engineering) of its applicability. While the results/outcomes appear reasonable, there could be various improvements.
Concerning the presentation, much of the work describing the process parameters, materials, etc. could be condensed into one or two tables. There appears to be an exorbitant number of "minor" sections that could be reasonably combined to form a few larger sections.
- Sections smaller than the 2rd sub were combined.
In terms of technical content, while the experimental and computational tools are described, the level of detail is sparse in many instances. Reproducibility by an outside practitioner would be very difficult, particularly in terms of the computational work. What governing equations were used, what were the initial/boundary conditions, what simplifying assumptions were used, etc.
- The governing equation (inherent strains) which Simufact Additive uses to simulate the shape distortion of the braking pedal was added. Besides, we added Figure 8 and discussed in more detail the boundary conditions and type of simulation that were used.
Besides, we changed the term Selective Laser Melting (SLM) to Powder Bed Fusion (PBF) to be more accurate following the standard ISO/ASTM 52900:2015.
Reviewer 3 Report
The current manuscript characterizes the properties of an additively manufactured motorcycle pedal part fabricated by selective laser melting. The title is attractive and interesting results were presented. However, there are some points that should be considered as follows:
- The main problem statement and novelty should be presented with more details in the introduction section.
- The main dimensions of the pedal part should be presented, the total height along the building direction is missing.
- A scale should be added to some figures, such as Figure 12.
- The results section should be titled "Results and discussion", more discussion is needed in this section.
- It is recommended to use the bullet points style in the conclusion section.
- There are some questions that should be answered through the manuscript as follows:
- What is the reference used for the applied SLM parameters?
- What is the effect of the anisotropic and nonhomogeneous microstructure of the additively manufactured part? did that point be taken during the simulation? is it recommended to do any postprocessing treatment of the as-built part?
- In general, the presented study is very interesting and it could be ready for publication after considering the above comments.
Author Response
Thank you for your constructive comments. We improved the paper accordingly and noted the changes as follows.
The current manuscript characterizes the properties of an additively manufactured motorcycle pedal part fabricated by selective laser melting. The title is attractive and interesting results were presented. However, there are some points that should be considered as follows:
The main problem statement and novelty should be presented with more details in the introduction section.
The main dimensions of the pedal part should be presented, the total height along the building direction is missing.
A scale should be added to some figures, such as Figure 12.
- The outer dimensions of 220x58x50 mm were mentioned in the technical drawing in Figure 9 and added in the discussion about the original pedal in Figure 3.
The results section should be titled "Results and discussion", more discussion is needed in this section.
It is recommended to use the bullet points style in the conclusion section.
There are some questions that should be answered through the manuscript as follows:
What is the reference used for the applied SLM parameters?
- We purchased from the machine provider these sets of SLM parameters, which are unfortunately confidential.
What is the effect of the anisotropic and nonhomogeneous microstructure of the additively manufactured part? did that point be taken during the simulation? is it recommended to do any postprocessing treatment of the as-built part?
- Discussed in the choosing of printing direction. Simulation in Simufact was done in macro scale, thus, it does not take into account the effect of the microstructure. The main concern at this scale is the shape distortion, which is considered by the inherent strain values. Postprocessing includes cutting, support removal, surface treatment and machining for the hole feature, which are added in the description for Figure 12.
In general, the presented study is very interesting and it could be ready for publication after considering the above comments.
Besides, we changed the term Selective Laser Melting (SLM) to Powder Bed Fusion (PBF) to be more accurate following the standard ISO/ASTM 52900:2015.
Round 2
Reviewer 1 Report
1. Please pay attention to the correct application of units, such as 275x150x90mm (should be 275mmx150mmx90mm ).
2. In Abstract and Conclusion section, the conclusions should include the specific comparable datum between conventional technoledge and PBF.
Author Response
Thank you again for your constructive comments. We have improved the paper according to your suggestions.
Please pay attention to the correct application of units, such as 275x150x90mm (should be 275mmx150mmx90mm ).
>>The noting for dimensions was corrected.
In Abstract and Conclusion section, the conclusions should include the specific comparable datum between conventional technology and PBF.
>>Unfortunately, the original pedal is aged and is not in good condition for testing. This has been added in the descriptions of Figure 3. It is also the reason why we conducted the physical test only on the 3D-printed pedal. On the other hand, we succeeded in restoring the shape of the historical pedal and proving that the restored pedal was fully functional. These statements were added in the Abstract and Conclusions.
Reviewer 2 Report
Nice work!
Author Response
Thank you for your encouragement!
Reviewer 3 Report
The revised manuscript is improved, however, some major issues still need to be considered as follows:
- Data for the characterization for both fresh and recycled powder should be presented in the materials and methods section.
- How many samples were printed for the solid and lattice structure? please illustrate.
- The authors presented an interesting work regarding the design and simulation. However, more experimental data are needed, in addition, a fair comparison should be presented between the experimental data of solid, lattice, and the original pedal properties.
- The properties of the additively manufactured materials are significantly different from the traditionally used material of the same alloy. So, material characterization and its mechanical properties should be well investigated.
- Post-processing should be illustrated for obtaining the final product for improving both surface quality and microstructure characteristics.
- Please note that there is no academic contribution obtained by adding simulation data without experimental validation. So, some of the data might need to be supported or removed.
Author Response
Thank you very much for your constructive comments. Please refer to the answers below and the highlighted text in the paper for our improvement.
The revised manuscript is improved, however, some major issues still need to be considered as follows:
Data for the characterization for both fresh and recycled powder should be presented in the materials and methods section.
>>We have elaborated on the material characterization in the new section about materials (please refer to subsection 2.6).
How many samples were printed for the solid and lattice structure? please illustrate.
>> We printed three solid pedals (1 used for testing the maximum force from the driver, 1 was subjected to geometric inspection with 3D scanning and ESPI, 1 was post-processed) and one with the lattice structure. This was added in the end of the Introduction and elaborated in subsection 3.3. and 3.4.
The authors presented an interesting work regarding the design and simulation. However, more experimental data are needed, in addition, a fair comparison should be presented between the experimental data of solid, lattice, and the original pedal properties.
>>The original pedal is unfortunately not in good condition for strength test due to its aging. This is the reason why we decided to conduct the physical test on the 3D-printed solid pedal. Additionally, as discussed in the previous revision, the lattice pedal did not meet the quality control to be tested. Therefore, the experimental data of the three versions cannot be acquired and compared.
However, we succeeded in restoring the exterior look of the pedal. Moreover, from the material assessments and physical tests, we were able to prove that the restored solid pedal was fully functional. This information has been added in the Abstract and Conclusion.
The properties of the additively manufactured materials are significantly different from the traditionally used material of the same alloy. So, material characterization and its mechanical properties should be well investigated.
>>Thank you for this note. To make the paper more interesting for readers of Materials journal we also decided to expand the materials section with the strain rate influence experimental analysis on the specimens printed from a virgin powder (unpublished results). Literature has been extended with our previous studies on SS316L. They are referenced in the text of the subsection 2.6.
Post-processing should be illustrated for obtaining the final product for improving both surface quality and microstructure characteristics.
>>The post-processed pedal has been added in subsection 2.3. and described in figure 17.
Please note that there is no academic contribution obtained by adding simulation data without experimental validation. So, some of the data might need to be supported or removed.
>> As described before, and explained in the paper, we had the possibility to validate the solid version of the 3D printed pedal only. This version of pedal was experimentally validated. An explanation has been added to the Conclusion that we proposed “a potential approach” to upgrade the braking pedal in terms of weight saving with lattice structure. According to reviews of other reviewers we want to keep the subsection 3.6 in the manuscript.
Your comments have helped us to improve this paper a lot.
Thank you with best regards.